# Recurrent Fever with Oral Lesions in Egyptian Children: A Familial Mediterranean Fever Diagnosis Not to Be Missed

**DOI:** 10.3390/children9111654

**Published:** 2022-10-29

**Authors:** Ahmed Omran, Ahmed Abdelrahman, Yasmine Gabr Mohamed, Mohamed Osama Abdalla, Eman R. Abdel-Hamid, Samar Elfiky

**Affiliations:** 1Department of Pediatrics & Neonatology, Faculty of Medicine, Suez Canal University, Ismailia 41522, Egypt; 2Department of Clinical Pathology, Faculty of Medicine, Suez Canal University, Ismailia 41522, Egypt; 3Medical Genetics Unit, Department of Histology and Cell Biology, Faculty of Medicine, Suez Canal University, Ismailia 41522, Egypt

**Keywords:** FMF, MEFV, oral lesions, Egyptian children, Serum amyloid A

## Abstract

Objectives: the aim of this study was to describe the genetic and clinical features of familial Mediterranean fever (FMF) in a group of Egyptian children. Materials and methods: This cross-sectional observational study included 65 children diagnosed with FMF according to the (Eurofever/PRINTO) classification criteria. The complete blood count (CBC), and acute phase reactants such as Serum amyloid A (SAA), and C-reactive protein (CRP) were all measured during the febrile episode. Mutation analysis for the MEFV gene was carried out for all subjects. Results: A total of 65 patients with FMF were included in the study. The first clinical manifestation was recurrent fever in all patients. Recurrent oral lesions accompanied fever in 63% of cases, abdominal pain in 31%, and musculoskeletal pain in 6%. The mean SAA level was 162.5 ± 85.78 mg/L. MEFV mutations were detected in 56 patients (86%). Among these patients, 6 (10.7%) were homozygous, while 44 (78.6%) were heterozygous. The most frequently observed mutation was E148Q 24 (37.5%), followed by M694I 18 (32.1%), and V726A 13 (20.3%). Half of the patients with oral lesions were E148Q positive, however abdominal pain was found to be higher in the patients with the M694I mutation. Conclusion: Recurrent fever with oral lesions could be an important atypical presentation of FMF in Egyptian children that should not be ignored and/or missed.

## 1. Introduction

Periodic or recurrent fever syndromes are defined by three or more episodes of unexplained fever in a 6-month period, occurring at least a week apart [1]. Familial Mediterranean fever (FMF) is a monogenic autoinflammatory disease and the most common of the periodic fever syndromes. FMF primarily affects people originating from the eastern Mediterranean area, including the Arab, Turkish, Armenian, and Jewish populations [2].

The diagnosis of FMF is made on the basis of clinical findings and is supported by ethnic origin, family history, and genetic testing. The clinical hallmark of FMF is recurrent febrile attacks associated with serositis (pleuritis, peritonitis, or synovitis), which usually last 1–3 days and resolve spontaneously. FMF patients cannot describe a consistent triggering event. Nevertheless, intercurrent infections, emotional stress, vigorous exercise, exposure to cold, fat-rich foods, drugs, surgery, and menstruation have been associated with an attack in some patients [3].

Significant overlap exists between the presenting symptoms of periodic fever syndromes, making the diagnosis challenging [4,5]. Diversity in FMF clinical presentation is mostly attributable to genotypic variation [6]. As a result, studying the phenotype–genotype association in children with FMF is a critical area of research [7,8,9]. An accurate and comprehensive examination is required for the early identification and differential diagnosis of FMF cases in order to prevent misdiagnosing patients with milder or unusual signs of the disease [10].

Recurrent aphthous stomatitis and other oral manifestations are commonly observed in autoinflammatory disorders like FMF and PFAPA (periodic fever, aphthous stomatitis, pharyngitis, and adenitis), and dysregulation of cellular immunity is presumed to be the underlying etiology [11]. The prevalence of oral mucosal lesions in FMF is variable among different ethnicities; in the Turkish population, Ayaz et al. reported that 9% of children with FMF presented with oral lesions [7]. In an Italian Center, however, the incidence of aphthous stomatitis was 28.2% in more than 370 patients [12].

Current data from the literature reported a significant overlap between FMF and PFAPA which have characteristic oral lesions [13,14,15,16,17]. Moreover, Butbul Aviel and colleagues [18] demonstrated that PFAPA and FMF, two frequent autoinflammatory disorders, are closely correlated in individuals of Mediterranean descent [18]. It is important for clinicians to realize that the clinical manifestation of one disease might evolve into that of another. The link between PFAPA and FMF raises the question of whether the MEFV gene has a similar pathophysiology and genetic impact on PFAPA.

The most pathogenic mutations in cases of FMF are found in exon 10. M694V, M680I, V726A, M694I, and E148Q in exon 2 are quite common among ethnic groups living in the Mediterranean region [19]. Serum amyloid A (SAA) is an apolipoprotein and among the principal acute-phase proteins produced by the liver. It is significantly raised after FMF episodes and normalizes within one to two weeks [20].

In this study, we provided a clinical observation of Egyptian children with recurrent fever, oral lesions, and elevated SAA levels that were unrelated to PFAPA and highlighted the phenotypic–genotypic association with MEFV mutations.

## 2. Materials and Methods

### 2.1. Study Design and Patient Population

This was a cross-sectional observational study that included 75 Egyptian children who were referred to the pediatric immunology clinic in the Suez Canal University Hospital, Ismailia, Egypt with recurrent periodic attacks of fever and other clinical manifestations (including recurrent oral lesions, recurrent abdominal pain, and musculoskeletal pain) between June 2020 and January 2022. A total of 65 patients were diagnosed to have FMF according to the (Eurofever/PRINTO) classification criteria [21] (Table 1).

Ten patients were excluded from the study because they did not fulfill the diagnostic criteria and were investigated for other causes of recurrent fever.

Patients were subjected to detailed history taking, including age at onset of symptoms, age of diagnosis, gender, other affected siblings, affected parents, and features of attacks (frequency, duration, and clinical manifestations). We observed that a percentage of the children presented with recurrent fever and oral lesions, which are not the classical aphthous stomatitis that is presented in PFAPA (Figure 1A,B). The range of the oral lesions observed in our study is illustrated in (Figure 1C–F).

### 2.2. Laboratory Investigations

During the attack, the complete blood count (CBC), and acute-phase reactants such as SAA, and C-reactive protein (CRP) were all measured. Mutation analysis for the MEFV gene was carried out for all subjects.

Five mL of blood samples were drawn from each subject. Samples were collected in EDTA and plain tubes. EDTA samples were used for CBC in the Sysmex XN-550 automatic cell counter (Sysmex Corp., Kobe, Japan). After complete clotting, plain tube samples were centrifuged at 4000 rpm for 10 min at room temperature for serum separation. Serum samples were used for CRP measurement using an immunoturbidimetric test with the Roche Diagnostics Cobas6000 (Roche Diagnostics GmbH, Mannheim, Germany). SAA was measured using the human SAA ELISA kit (ab100635, Abcam, Cambridge, UK).

Analyses of The MEFV gene mutations were carried out using the FMF Strip Assay ^TM^ (Vienna Lab Labor Diagnostika GmbH, Vienna, Austria) according to the manufacturer’s protocol. Briefly, genomic DNA isolation was carried out from EDTA blood, followed by PCR amplification using biotinylated primers. Purified amplification products were then hybridized to a test strip containing mutant allele-specific oligonucleotide probes immobilized as an array of parallel lines. Bound biotinylated sequences were detected using streptavidin–alkaline phosphatase and color substrates. The assay covers 12 common mutations in the MEFV gene: E148Q, P369S, F479L, M680I (G/C), M680I (G/A), I692del, M694V, M694I, K695R, V726A, A744S, and R761H.

### 2.3. Statistical Analysis

Data were analyzed using the statistical package for social sciences (SPSS) for windows version 25.0 (SPSS, Chicago, IL, USA). Descriptive data were presented as mean ± SD or percentages. We used the analysis of variance (ANOVA) test to detect statistically significant differences between two or more categorical groups by testing for differences of means.

## 3. Results

A total of 75 patients presented with recurrent fever and other clinical manifestations, including recurrent oral lesions, abdominal pain, and musculoskeletal pain, of which 65 (38 males and 27 females) were diagnosed with FMF. The age of disease onset ranged from 10 months to 10 years. Positive family history was observed in 21 (32.3%) patients. Positive heterozygous V726A was found in 9 patients, 3 of whom were related, and was followed by positive heterozygous E148Q in 6 patients. Recurrent fever was the most frequent clinical presentation, observed in 65 (100%) of the cases, followed by oral lesions in 41 (63%), followed by abdominal pain, observed in 20 (31%). The least frequent clinical presentation in our study was musculoskeletal pain, observed only in 4 (6%). The SAA mean was 162.5 ± 85.78 mg/L. MEFV mutation was observed in 56 (86%). A homozygous mutation was found in 6 (10.7%) patients, a compound heterozygous mutation was found in 6 (10.7) patients and a heterozygous mutation was found in 44 (78.6%) patients. The most prevalent four mutations were E148Q, M694I, V726A, and P369S. M694I was the most common mutation in homozygous individuals (46.67%). In compound heterozygotes, E148Qand P369S were presented with each other in 5 (83.3%) patients. Table 2 summarizes this data.

In 41 cases (63.1%) who presented with recurrent oral lesions and a high SAA, 20 (48.8%) of them were E148Q positive. Of the 20 patients who presented with recurrent abdominal pain and a high SAA, 90% of them were M694I positive. Only four patients presented with recurrent musculoskeletal pain, and all of them were V726A positive.

In our study, SAA levels showed a statistically significant difference with the MEFV genetic mutation which was highest with the M6941 gene (211.8 ± 46.1 mg/L) compared with less aggressive mutations such as E148Q (170.9 ± 79.9 mg/L). SAA showed no statistically significant differences relative to mutation type Table 3.

## 4. Discussion

FMF is an autosomal recessive autoinflammatory disorder with a specific predilection for Mediterranean races. As part of the heterogeneous phenotypic presentation of FMF in children, we documented a clinical observation of recurrent fever with recurrent oral lesions as the initial symptoms in Egyptian children with FMF.

We found that 95.4% of patients in our study had their first symptoms appear in the first 5 years of life. However, diagnostic delays are common, and an adult and late-onset FMF which is defined as a disease onset ≥40 years is described in patients with mild forms [20,22,23]. This is consistent with the findings of other studies, which show that the clinical onset and diagnosis of FMF typically occur during the first decade of life, and most often between the ages of 1 and 5 years.

Positive family history was reported in 21 (32.3%) patients, which may be attributed to the recessive inheritance pattern of the disease. Furthermore, positive consanguinity increases the frequency of homozygotes in the Egyptian population, which results in a higher-than-expected FMF incidence. Consistent with our findings, positive family history in previous studies varied from 20% to 46.1% [7,24].

In the present study, recurrent fever was the most frequent clinical manifestation presented in all patients. Fever was accompanied by recurrent oral lesions in 63%, abdominal pain in 31%, and musculoskeletal pain in 6%, however, none of the patients presented with chest pain or had erysipelas-like erythema. In contrast to our findings, Çakan, et al. [20] reported that the clinical features in FMF attacks were fever (100%), abdominal pain (96.4%), chest pain (42.8%), and arthritis (25%). Another large Turkish cohort of 3454 patients showed that 88.2% had abdominal pain, 86.7% had a fever, 27.7% had arthritis, 20.2% had chest pain, 23% had myalgia, and 13.1% had erysipelas-like erythema [25].The phenotypic variations among and within different ethnic groups could be explained by a variable MEFV mutation pattern, modifier genes, or environmental and population-specific factors such as demographic history, as well as a complex combination of climatic and geographic features [26].

Our results showed that the most frequently reported MEFV mutations were E148Q, M694I, and V726A and were detected in 37.5%, 32.1%, and 20.3% of FMF patients, respectively. Several studies of MEFV mutations in different ethnic groups between 2007 and 2021 reflect that the E148Q mutation is more common in Egypt than the M694V variant, unlike other ethnic groups [27,28]. In an Egyptian study, Talaat et al. evaluated 70 patients with FMF; they demonstrated that E148Q, V726A, and M680I were the most common mutations and were detected in 20%, 15.7%, and 14.3% of patients, respectively [24]. Another Egyptian study reported that E148Q is the most frequent mutation [29]. In the Mediterranean region, FMF is very common, and the mutation frequency shows marked variability. In addition, the whole spectrum of genetic variations that manifest as clinical FMF has not been elucidated. This might explain the clinical diversity and variable disease severity among different ethnicities [30].

We found that most of our patients (63.1%) presented with recurrent fever and oral lesions like ulcers, blisters on the tongue, and an erythematous palate.

Despite the fact that recurrent aphthous mouth ulcers are commonly reported in other autoinflammatory disorders, such as PFAPA syndrome and mevalonate kinase deficiency [31,32], Patients with FMF were also observed to experience oral lesions. In their study, KonéPaut et al. [33] reported that 21% of patients developed recurrent oral ulcers as well as a significant incidence of mucocutaneous features. Furthermore, Manna et al. [12] and Li et al. [34] have also reported similar results in both children and adult populations. Another study conducted by Esmeray and colleagues [11] evaluated the oral health status of 199 children with FMF. They demonstrated that recurrent aphthous mouth ulcers occurred in 77 (38.7%) of the patients. It is hypothesized that cellular immunity disruption is the underlying cause of recurring aphthous mouth ulcers. Colchicine binds to microtubular proteins and inhibits the migration and phagocytosis of granulocytes. Therefore, it may be useful in treating aphthous mouth ulcers.

An oral aphthous ulcer was thought to be a significant component of PFAPA syndrome rather than FMF. However, in Mediterranean countries, FMF patients may present with a PFAPA-like phenotype at early ages [18]. Recent studies show that there are many common features between these two distinct diseases, despite their different etiologies [15]. Moreover, the co-occurrence of PFAPA syndrome and FMF was observed in more than one-third of Armenian FMF children. They had a higher frequency of the MEFV compound heterozygous genotype and developed FMF at an earlier age. The condition was also more severe, with more frequent episodes of fever [35]. According to Celiksoy et al. [36], the majority of patients with PFAPA syndrome had heterozygous MEFV gene mutations. Therefore, further study is required to determine if carrying a heterozygous MEFV gene is the primary cause of this condition.

In our study, the E148Q mutation was the most prevalent in patients presenting with oral lesions (48.8%) and the most common in the whole study. Li et al. reported in their case series that E148Q mutation was found in all patients presented with oral ulcers (18.2%) [34]. Whether the E148Q mutation is a disease-causing mutation, or polymorphism is still a matter of debate. Other researchers supported our results that heterozygous mutations for E148Q had an important role in the pathophysiology and etiology of FMF disease and were associated with a milder disease course [37,38].

In the present study, recurrent abdominal pain was the second most common clinical presentation, occurring in 20 patients (31%). Contrary to these results, a large Egyptian cohort revealed that abdominal pain was observed in 90.8% of cases. Among these patients, the most frequent genotype was M694V/V726A (37.7%) [39]. Peritonitis was observed in 92.3% of patients carrying M694I, which was significantly higher than in the patients carrying E148Q [5]. In an Arabic group of patients, the most common signs observed were peritonitis (93.7%), arthritis (33.7%), and pleurisy (32%) [22]. Similarly, in a mixed population of Sephardic Jews, Armenians, Arabs, Turks, French, and others, the main clinical characteristics of the patients were fever (83.3%), abdominal signs (74.0%), thoracic signs (24%), joint signs (50.3%), erysipelas-like erythema (8.2%), splenomegaly (8.5%), and amyloidosis (4%) [5]. In research that was conducted in Turkey, abdominal pain (76%) and fever (58%) were the two most common manifestations among patients, followed by arthritis (28%) and chest pain (19%) [39]. Another Turkish study detected MEFV gene mutations in forty-three patients with epigastric pain syndrome (57.3%) and the carrier rate was 30.0%. The most common MEFV gene mutation was R202Q (55.8%), followed by E148Q (16.2%), R761H (16.2%), V726A (9.3%), M680I (9.3%), and M694V (4.6%) [40]. These different findings may be explained by the small number of younger patients in our study. Additionally, the difference in racial groups and geographic regions may be due to the difference in the predominant gene mutation in different populations (phenotype–genotype correlation).

The genetic studies of 9 (13.8%) of our patients with the classical clinical presentation were negative for any of the 12 common MEFV-screened mutations. Nevertheless, genetic analyses are useful in confirming a diagnosis of FMF and identifying atypical FMF presentations as well as in identifying pre-symptomatic patients’ relatives [41]. Many studies have shown that between 10% and 20% of patients with FMF do not have mutations in the MEFV gene [42,43]. Recent evidence has shown that the focused next-generation sequencing (NGS) approach is crucial for detecting all mutation types (new, uncommon, and known) linked with FMF. In these studies, NGS was used to discover 56 unique mutations, 141 different genotypes, and 2 novel variants. About 46 of the newly discovered mutations are extremely uncommon. This approach also helps to improve medication therapy for FMF by elucidating the genetic profile of individuals with atypical phenotypes [44,45,46,47].

In this study, SAA levels were increased in 100% of the total cohort and showed significantly elevated levels relative to the number of mutations, the highest being with complex mutation, and it also showed a statistically significant correlation with the type of mutations, the highest being with the *M6941* mutation. Consistent with our findings, Çakan et al. [20] reported that 100% of their patients had elevated SAA levels and that the cutoff value for distinguishing FMF attacks from febrile infection was 111.5 mg/L, with a sensitivity of 100% and a specificity of 65.1%.

Based on our findings, assessing SAA levels can be a valuable aspect of the diagnostic assessment in children with recurrent fever if there is no indication of a current infection, and it might be a very important marker for moving forward with genetic testing in individuals with a high suspicion of FMF.

## 5. Conclusions

Recurrent fever with oral lesions could be an important atypical presentation of FMF in Egyptian children that should not be ignored and/or missed. In low-resource countries, SAA could be a very promising marker during an FMF attack in children to precede *MEFV* genetic analysis. Genotypic–phenotypic relationships in children with FMF still need further investigation.

## Figures and Tables

**Figure 1 children-09-01654-f001:**
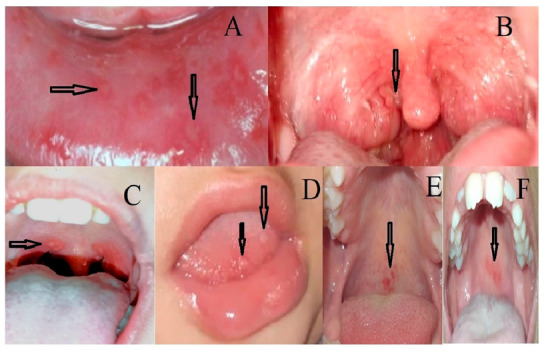
(**A**,**B**) the classical aphthous stomatitis and pharyngitis in PFAPA. (**C**–**F**) the range of oral lesions presented in our patients. (**C**) Ulcerative lesions in the soft palate and posterior oropharynx. (**D**) Blisters in the tongue tip. (**E**,**F**) Erythematous palate.

**Table 1 children-09-01654-t001:** Eurofever/PRINTO classification criteria for FMF.

Eurofever/PRINTO Classification Criteria
Presence of confirmatory MEFV genotype * and at least one among the following: ● Duration of episodes 1–3 days. ● Arthritis ● Chest pain ● Abdominal pain
Or
Presence of not confirmatory MEFV genotype ^†^ and at least two among the following: ● Duration of episodes 1–3 days. ● Arthritis ● Chest pain. ● Abdominal pain

* Homozygous or biallelic compound heterozygous for pathogenic or probably pathogenic mutations; ^†^ Compound heterozygous for 1 pathogenic variant and 1 VUS, or biallelic VUS, or heterozygous for 1 pathogenic MEFV variant.

**Table 2 children-09-01654-t002:** Patient’s basic characteristics.

Demographic Data *n* (%)	
Age range of diagnosis Mean ± SD (Years)	(10 months–10 years) 3.5 ± 1.55805
Male	38 (58.5%)
Female	27 (42.5%)
Positive family history	21 (32.3%)
**Main clinical presentations *n* (%)**	
Recurrent Fever	65 (100%)
Recurrent oral lesions	41 (63%)
Recurrent abdominal pain	20 (31%)
Recurrent musculoskeletal pain	4 (6%)
**Laboratory data**	
SAA level Mean ± SD (mg/L)	162.5 ± 85.78
CRP (mg/dL)	33 ± 15
**Genetic data (Type of mutations) *n* (%)**	
Homozygous	6 (10.7%)
Heterozygous	44 (78.6%)
Compound Heterozygous	6 (10.7%)
**Genetic data (Common mutations) *n* (%)** 64 mutations in 56 patients	
E148Q	24 (37.5%)
M694I	18 (32.1%)
V726A	13 (20.3%)
P369S	5 (7.8%)
M680I(G/A)	3 (4.7%)
R761H	1 (1.6%)

**Table 3 children-09-01654-t003:** Comparison between SAA levels and mutation type, and different genes.

Mutation Type	SAA Mean ± SD	MEFVMutation	SAA Mean ± SD
Homozygous	193.1 ± 20.61899	E148Q	170.9 ± 79.9
Heterozygous	181.8 ± 74.1403	M680I	205.0 ± 8.2
Combined Heterozygous	194.7 ± 16.01879	M694I	211.8 ± 46.1
*p*-value	0.849	V726A	178.6 ± 20.1
		*p*-value	0.003

## Data Availability

Datasets are available on request.

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
