# Peer review of "Recurrent Fever with Oral Lesions in Egyptian Children: A Familial Mediterranean Fever Diagnosis Not to Be Missed"

_children, 2022, doi:10.3390/children9111654_

Round 1
Reviewer 1 Report
The authors present an interesting manuscript on the clinical and laboratory diagnostic as well as molecular genetic workup of 65 children with FMF expanding on oral lesions as an important atypical presentation of the disease.
Although the manuscript has its merits it may not be published in the current version as it needs major revision.
Comments:
Page 1 , line 34: syndromes. FMF primarily affects………
Page 2, line 44: making the diagnosis challenging
Page 2, line 48:……………FMF cases in order to prevent misdiagnosing patients with….
Page 4, line 125: were diagnosed with FMF and were male /female
Page 4, line 126; …ranged from 10 months…
Page 4, line 127-128 : please avoid expression like positive markers aso throughout the manuscript. Rather refer to a patients as being heterozygous for ……
Page 4 line130-131: You metion that recurrent fever was present in all cases, so there is no need to mention it with other symptoms.
Page 4 line 134: Please replace genetic with MEFV
Page 4 line 135: What is the difference between a single mutation and a heterozygous mutation? Please always refer to heterozygous/homozygous/compound herterozygous/complex mutations. Do not refer to a single, double triple mutation.
Page 4 line 237: please omit subsequently.
Pages 4/5, Table 2: Please omit recurrent fever with oral lesion aso , please just state oral lesions aso.
Furthermore, please omit number of mutations in the table. Please always refer to heterozygous/homozygous/compound herterozygous/complex mutations.
Moreover, replace Q761H with R761H
Figure 2 should be omitted since it does not contain additional information and is rather confusing. Consequently, also omit the sentence in line 141.
Page 5, lines 144/ 145: Please avoid M694I positive aso. Please refer to as M694I heterozygotes aso. Throughout the manuscript and for all mutations.
Page 6, line 150. Being highest in patients with complex mutations , not triple mutations.
Page 6, line 152, what is a wild mutation?
Page 6, Table 3: Please replace Genetic with MEFV mutation and omit column listing mutations numbers. It is sufficient to report on heterozygous/homozygous/compound herterozygous/complex mutations.
Page 6, line 162: Please be more specific concerning adult and late onset FMF (please refer to Kriegshäuser et al., 2018.
Page 6; line 172: What is the first clinical manifestation? Did you record ii first or do you mean that this clinical manifestation is the most frequent one. Please be more specific.
Page 6, lien 177: Another large….
Page 6, line 186, Please be mor consistent throughout your manuscript, either using variant odr mutation. I would suggest to stick to mutation rath than to variant.
Page 7, lines 220-221. The authors report E148Q to be most frequent in patients with oral lesions. Are there other studies on this matter? Please discuss in more detail.
Page 7 line 226: please omit recurrent fever as this is the most frequent clinical sign.
Page 7 line 243: Please replace with……..by the small number of younger patients in our study.
Page 8: line264: It is common knowledge that SAA is found elevated in patients with FMF and )therefore I am not quite sure, why the authors included this reference (Lofty et al.)? Is there a special reason for it? If not, please omit this reference.
Page 8: 274: genotype-phenotype relationships ---- still need further investigation.
Author Response
Reviewer comments:
The authors present an interesting manuscript on the clinical and laboratory diagnostic as well as molecular genetic workup of 65 children with FMF expanding on oral lesions as an important atypical presentation of the disease.
Although the manuscript has its merits it may not be published in the current version as it needs major revision.
Author’s response:
Response:
Thank you for your valuable time in reviewing our manuscript. We appreciate your positive comments. We believe that correcting our manuscript according to your kind suggestion markedly improved it.
1) Reviewer editing comments:
Page 1 , line 34: syndromes. FMF primarily affects………
Page 2, line 44: making the diagnosis challenging
Page 2, line 48:……………FMF cases in order to prevent misdiagnosing patients with….
Page 4, line 125: were diagnosed with FMF and were male /female
Page 4, line 126; …ranged from 10 months…
Page 4, line 127-128 : please avoid expression like positive markers aso throughout the manuscript. Rather refer to a patients as being heterozygous for ……
Page 4 line130-131: You metion that recurrent fever was present in all cases, so there is no need to mention it with other symptoms.
Page 4 line 134: Please replace genetic with MEFV
Page 4 line 135: What is the difference between a single mutation and a heterozygous mutation? Please always refer to heterozygous/homozygous/compound herterozygous/complex mutations. Do not refer to a single, double triple mutation.
Page 4 line 237: please omit subsequently.
Page 5, lines 144/ 145: Please avoid M694I positive aso. Please refer to as M694I heterozygotes aso. Throughout the manuscript and for all mutations.
Page 6, line 150. Being highest in patients with complex mutations , not triple mutations.
Page 6, Table 3: Please replace Genetic with MEFV mutation and omit column listing mutations numbers. It is sufficient to report on heterozygous/homozygous/compound herterozygous/complex mutations.
Page 6; line 172: What is the first clinical manifestation? Did you record ii first or do you mean that this clinical manifestation is the most frequent one. Please be more specific.
Page 6, lien 177: Another large….
Page 6, line 186, Please be mor consistent throughout your manuscript, either using variant odr mutation. I would suggest to stick to mutation rath than to variant.
Page 7 line 226: please omit recurrent fever as this is the most frequent clinical sign.
Page 7 line 243: Please replace with……..by the small number of younger patients in our study.
Page 8: 274: genotype-phenotype relationships ---- still need further investigation.
1) Response: We edited all the reviewer suggestions all over the manuscript.
2) Reviewer comments: Pages 4/5, Table 2: Please omit recurrent fever with oral lesion also , please just state oral lesions also. Furthermore, please omit number of mutations in the table. Please always refer to heterozygous/homozygous/compound herterozygous/complex mutations. Moreover, replace Q761H with R761H
2) Response: we edited table 2 according to the reviewer suggestions
3) Reviewer comment: Figure 2 should be omitted since it does not contain additional information and is rather confusing. Consequently, also omit the sentence in line 141.
3) Response: we omitted figure 2 according to the reviewer kind suggestions.
4) Reviewer comment: Page 6, line 152, what is a wild mutation?
4) Response: After extensive search we didn’t find a constant definition for the wild mutation in the FMF gene. Although it mentioned in many published researches but we preferred to omit it from the whole manuscript to avoid misleading of the readers.
5) Reviewer comment: Page 6, line 162: Please be more specific concerning adult and late onset FMF (please refer to Kriegshäuser et al., 2018.
5) Response: We added these data in the discussion part.
Adult and Late-onset FMF is rare and has higher levels of clinical and genetic variability than previously thought [24].
6) Reviewer comment: Page 7, lines 220-221. The authors report E148Q to be most frequent in patients with oral lesions. Are there other studies on this matter? Please discuss in more detail.
6) Response: We discussed the only available data from Li et al case series.
Li et al reported in their cases series that E148Q mutation was found in all patients presented with oral ulcers [35].
7) Reviewer comment: Page 8: line264: It is common knowledge that SAA is found elevated in patients with FMF and ) therefore I am not quite sure, why the authors included this reference (Lofty et al.)? Is there a special reason for it? If not, please omit this reference.
7) Response: we omitted the reference.
Reviewer 2 Report
I read with a great interest the article entitled Recurrent fever with oral lesions in Egyptian children: a FMF diagnosis not to be missed by Omran et al
This research article was about genetic and clinical features of FMF in 65 Egyptian children.
The main finding was the presence of recurrent oral lesion in patients with heterozygous mutation and in presence of E148Q variant which is not considered as pathogenic variant for FMF actually as E148Q is common in Arabic population. But this finding still have a good interest because in our practice recurrent oral lesion is a common symptoms and criteria for Behçet disease are not always fulfilled. Maybe further research should be performed in the future for patient with this phenotype and genotype.
The main message from this research article is that we should think about FMF in patient with recurrent oral lesion with fever in patients from the Mediterranean region.
I have a request for the author concerning table 3, the last column I did not understand the relevance of this data
Author Response
Reviewer comments
I read with a great interest the article entitled Recurrent fever with oral lesions in Egyptian children: a FMF diagnosis not to be missed by Omran et al
This research article was about genetic and clinical features of FMF in 65 Egyptian children.
The main finding was the presence of recurrent oral lesion in patients with heterozygous mutation and in presence of E148Q variant which is not considered as pathogenic variant for FMF actually as E148Q is common in Arabic population. But this finding still have a good interest because in our practice recurrent oral lesion is a common symptoms and criteria for Behçet disease are not always fulfilled. Maybe further research should be performed in the future for patient with this phenotype and genotype.
The main message from this research article is that we should think about FMF in patient with recurrent oral lesion with fever in patients from the Mediterranean region.
Response:
Thank you for your valuable time in reviewing our manuscript. We appreciate your positive comments. We believe that correcting our manuscript according to your kind suggestion markedly improved it.
Reviewer comment: I have a request for the author concerning table 3, the last column I did not understand the relevance of this data
Response: We found that the level of SAA is higher with more aggressive MEFV mutations like M680I and M694I compared with milder mutation like E148Q. We think that these data could help pediatricians who are not familiar with reading the SAA results to correlate between the type of mutation and SAA levels.
Round 2
Reviewer 1 Report
The manuscript has improved significantly, however, I would like to address a couple of points:
Page 4 (lines 131-132): …, of which 65 (38 males and 27 females) were diagnosed with FMF.
Page 4 (line 134): Please omit : of which only 2 patients have positive mothers
Page 6 (lines 180-183). I did not want the authors to cite the work by Kriegshäuser et al, just for the sake to include it. It was rather meant to clarify the definitions of adult and late-onset. Here the authors modify – for example late-onset disease is defined as a disease onset ≥ 40 years.
Please omit: Adult and Late-onset181
FMF is rare and has higher levels of clinical and genetic variability than previously182
thought
Page 6/7 (lines 285-286): In the study of Li et al, what was the percentage of patients with oral ulcers.
Page 7 (line 308): … of younger of younger patients? This does not make sense.
Author Response
Reviewer Comments: The manuscript has improved significantly, however, I would like to address a couple of points:
Authors response:
Thank you so much for your kind revision and suggestions. We corrected all listed minor corrections and we hope we can get your final approval.
Page 4 (lines 131-132): …, of which 65 (38 males and 27 females) were diagnosed with FMF.
Page 4 (line 134): Please omit : of which only 2 patients have positive mothers
Page 6 (lines 180-183). I did not want the authors to cite the work by Kriegshäuser et al, just for the sake to include it. It was rather meant to clarify the definitions of adult and late-onset. Here the authors modify – for example late-onset disease is defined as a disease onset ≥ 40 years.
Please omit: Adult and Late-onset181
FMF is rare and has higher levels of clinical and genetic variability than previously182
thought
Page 6/7 (lines 285-286): In the study of Li et al, what was the percentage of patients with oral ulcers.
Page 7 (line 308): … of younger of younger patients? This does not make sense.